# The Combination of School-Based and Family-Based Interventions Appears Effective in Reducing the Consumption of Sugar-Sweetened Beverages, a Randomized Controlled Trial among Chinese Schoolchildren

**DOI:** 10.3390/nu14040833

**Published:** 2022-02-16

**Authors:** Chenchen Wang, Xin Hong, Weiwei Wang, Hairong Zhou, Jie Wu, Hao Xu, Nan Zhou, Jinkou Zhao

**Affiliations:** 1Department of Non-Communicable Disease Prevention, Nanjing Municipal Center for Disease Control and Prevention, Nanjing 210003, China; isisccwang@163.com (C.W.); nj_hongxin@126.com (X.H.); nj_wangww@126.com (W.W.); zhouhrong@126.com (H.Z.); wujie_0616@126.com (J.W.); 18375314830@163.com (H.X.); 2Department of Non-Communicable Disease Prevention, Jiangsu Provincial Center for Disease Control and Prevention, Nanjing 210009, China

**Keywords:** sugar sweetened beverages, student, school, family, intervention, randomized controlled trial

## Abstract

To examine whether environmental interventions, student awareness and parents’ model roles are associated with the consumption of sugar-sweetened beverages (SSBs), a randomized controlled trial was conducted among Chinese schoolchildren. A multi-stage cluster random sampling method was applied to select four primary schools, two in urban areas and two in rural areas, in Nanjing, eastern China. Classes of the third grade in the selected four schools were randomly assigned to the intervention group and control group. Among selected students in those classes, aged 9–10 years, those in the intervention group received intervention measures comprising school-based and family-based measures and accepted monthly monitoring along with interventions, for two consecutive semesters, while those in the control group did not receive any specific interventions. After intervention, there was a significant increase in SSB knowledge and an improvement in the family environment with parents in the intervention group. The proportion of frequent consumption (≥4 times/week) of any SSBs in the intervention group was lower than that in the control group (31.5% vs. 56.2%, *p* < 0.01). Multivariate analysis indicated that parental education level is positively associated with reduced SSB consumption. Interventions showed an average decrease in SSBs consumption by 1.77 units, those living in urban areas decreased by 2.05 units. The combination of school-based and family-based interventions appears effective in reducing SSB consumption among Chinese schoolchildren, especially in urban areas and for those with parents with lower educational levels.

## 1. Introduction

Sugar-sweetened beverages (SSBs) are liquids sweetened with various forms of additive sugars. SSBs include a range of soft drinks, from carbonated beverages, fruit and/or vegetable beverages, sweetened tea, to sports drinks and lactobacillus or milk beverages [1]. The additive sugars include brown sugar, corn sweetener, corn syrup, glucose, high-fructose corn syrup, honey, lactose, malt syrup, and sucrose [2]. Such additive sugars are sources of high energy that have poor nutritional value and are considered the primary source of sugar intake in children’s diets [3,4].

During the past decades, the consumption of SSBs has risen globally in both developed and developing countries. From 1977 to 2002, the consumption of SSBs increased from 15% to 33% as a percentage of total beverage intake by children aged 6 to 11 years in the United States [5]. SSBs account for 14% of total energy intake among children aged 4–18 years in the United Kingdom [6], and 10% among children aged 6–11 years in Mexico [7].

Similarly in China, the consumption of SSBs among children increased, along with the increased production and sales of beverages [8,9,10]. The China Nutrition and Health Survey conducted from 2010 to 2012 showed 61.9% children aged 6–17 years drinking SSBs at least once a week, compared with 4.7% children aged 6–13 years drinking SSBs at least once every 3 days in 2002 [11]. In 2013, a cross-sectional survey with a total number of 10,091 of students of the fourth grade (aged 10–11 years) and the seventh grade (aged 13–14 years) shows that 48.5% of all students had an experience of consuming SSBs at least once a week in Nanjing [12], the capital of Jiangsu province in eastern China. 

The increase in SSBs intake is one of several dietary changes that have been attributed to the nutrition transition. SSBs provide so-called empty calories [13], with less feeling of fullness or satiety than solid food [14]. Compelling evidence supports associations of SSBs intake with increased energy intake, and their important contribution to obesity and obesity-related diseases [15,16]. The high intake of SSBs among children is also associated with lower intakes of water, milk, fruits, and vegetables [17] and may lead to an increased risk of several medical problems, including lower micronutrient status and increased risk of caries [7,12,18,19,20,21,22]. The increase in SSBs intake has become an increasingly visible public health and policy issue [7,20].

Public health interventions to reduce SSBs consumption in children have been increasingly available over the past two decades [23]. Interventional efforts, to limit SSBs consumption among children and adolescents, focused on school only or family only. The school intervention, including educational and environmental components, provided to students by trained teachers, had no significant effect on soft drink consumption in intervention or control groups [24,25,26,27]. The home-based intervention, including a parental involvement component and counseling sessions offered to parents, showed no significant difference between intervention and control groups [6,28,29]. Most previous interventions adopted educational and behavioral approaches that focus on improving the knowledge, attitude, and, subsequently, behavior of children toward SSBs, and ignore patients’ role models and environmental aspects. Based on existing studies, there are still inconclusive evidence of what effective interventions are [23,24].

Without environmental changes, effective behavioral interventions targeted at individuals may not sustain behavioral change [30]. Children, unlike adults, have no flexibility to choose the environment to live in or to drink beverages [31]. Schools, where children spend 8 or more hours daily during weekdays, may play important roles in reducing their SSB consumption [25,26]. However, there are few relevant studies targeting SSB consumption in children that incorporate both students and parents education and a supportive healthy school and home environment in promoting students to take in fewer SSBs globally.

The main purpose of this randomized controlled trial is to test the hypothesis that the combination of school-based and home-based environmental interventions, student awareness, and parents’ role models are positively associated with reduced SSB consumption among children.

## 2. Subjects and Methods

### 2.1. Study Subjects and Selection

The study selects students in the third grade of primary school, aged 9–10 years, using a multi-stage cluster sampling method. All administrative districts in Nanjing, China, were firstly divided into urban and rural strata. One district was then randomly chosen from rural and urban strata respectively. In each chosen district, two primary schools were selected at random, respectively, from the roster of primary schools in the district. All students in the third grade in selected schools were invited to participate in the study.

### 2.2. Inclusion and Exclusion Criteria

Subjects are eligible for the study, when they were free of any medical conditions that would limit their ability to change beverage consumption behaviors, currently enrolled in the third grade with parental or caregiver’s consent to participate. The exclusion criteria include having reported birth defects including congenital disease, hydrocephalus of deformity at birth, or having abnormal mental behavior, or lack of parental or caregiver’s consent for participation.

### 2.3. Ethics Statement

Written informed consent was obtained from a parent or a caregiver. The study protocol was ethically cleared by the Academic and Ethical Committee of Nanjing Municipal Center for Disease Control and Prevention (Approval No. 2019-002) and registered at the Chinese Clinical Trial Registry (Test registration No. ChiCTR2000033945).

### 2.4. Study Design

This cluster randomized controlled trial (RCT) was conducted among selected primary schools in Nanjing, China, where schools served as clusters. Two schools in either rural or urban strata were randomly assigned into the intervention group or control group, respectively.

The primary outcome variable was the change in the proportion of infrequent (<1 time/week) consumption of SSBs. Assuming a difference of at least 8% of SSBs intake frequency before and after the intervention [32], 1 − β = 0.90, a power of 90% and a level of significance of 5%, considering a dropout rate of 20%, the minimum sample size would be 690 for intervention and control groups, respectively. 

Prior to the intervention, the baseline assessment was performed using a standard questionnaire, self-administered by selected students. The questionnaire included date of birth, sex, name of the school, school grade, school class, knowledge about SSBs, family environment regarding SSBs and the weekly frequency of SSB consumption. Parental education level, body weight and height were self-reported by parents when obtaining their informed consent.

Students in the intervention group agreed to receive intervention measures and accepted monthly monitoring along with interventions, for two consecutive semesters. Those in the control group accepted the regular monitoring, once per semester, without any interventions. The outcome assessment was conducted at 12 months after baseline using exactly the same questionnaire as that for baseline assessment.

### 2.5. Knowledge about Sugar-Sweetened Beverages (SSBs)

General knowledge about SSBs was measured using 10 questions, correct answers of which are derived from eight health education courses. Correct answers were recoded as 1, wrong answers or ‘I don’t know’ answers as 0. Those with a score of 6 or higher were classified as ‘adequate knowledge about SSBs’. 

### 2.6. Family Environment of SSBs

A questionnaire with six statements was used to measure family environment regarding SSBs, including family beverage drinking environment, parent’s attitude, and behavior toward SSBs. Parents select either “disagree” or “agree” against each statement.

### 2.7. The Weekly SSBs Intake Frequency

The weekly SSBs intake frequency questionnaire was administrated to obtain information on the children’s consumption of beverages. Each beverage was categorized based on nutrient content and on China’s Beverage General Rule (GB10789-2008). Eight broad beverage categories were used, namely carbonated beverages, fruit and/or vegetable beverages, sweetened tea beverages, lactobacillus milk beverages, sports and energy beverages, plant-protein beverages, brewed beverages and coffee beverages. The consumption of SSBs was measured on a 7-day frequency scale, using the question ‘How many times did you drink carbonated beverages in past week, commonly available in the market?’. The frequency of SSBs consumption was calculated when either of eight categories of SSBs consumption was reported.

### 2.8. Intervention

The intervention measures comprise school-based and family-based measures. 

There were four components in the school-based interventions. A. Health education courses, compulsory as part of the curriculum, a 15-min session delivered monthly as a video, administrated in the classroom by the health care teacher. The courses teach (a) knowledge of SSBs, such as definition and types of SSBs, how to read nutritional labels on the packages and how to choose the right beverage; and (b) the harm caused by SSBs, such as dental caries, overweight and obesity, osteoporosis, and increased risk of chronic diseases in adulthood. B. School environment support includes eight posters with different themes of SSBs posted in classroom, gymnasium, canteen, corridor, playground and other public places. C. Class environment support focuses on the class bulletin board updated monthly by students, with content in line with monthly 15-min classroom session. D. Sugar-free school campaign includes free drinking water available for all students on campus and prohibited sales of SSBs on campus.

The family-intervention includes four components. A. Health lectures, once per semester, were offered to the parents by the study principal investigator and research assistants to promote reduced SSBs consumption. B. Instant core messages were delivered monthly using social media channels, such as parent WeChat group managed by the class advisor and self-supported parents QQ group including during the summer and winter school breaks. The message was prepared by the study team, in accordance with contents of in-classroom health education sessions. C. Little Hands holding Big Hands. Each of students received a pictorial intervention booklet, brightly colored with key messages, developed by the study team. Students (Little Hands) take the booklet home and have interactive activities with their parents (Big Hands). D. Student–parent paired collaboration. Each semester, students and parents collaborated to create paintings and tabloids themed ‘Reduce SSBs and Enjoy a Healthy Life’. 

### 2.9. Quality Control

Pre-testing was done on 32 individual subjects in the third grade in a school who are not included in the study, to ensure participants understand the questionnaire. In each school, the study team comprised staff from the district center for disease control and prevention (CDC), a class teacher and a health care teacher. They were trained by the principal investigator on how to conduct a questionnaire survey visually with a standard PowerPoint slide set in each class. The questionnaire filled by students was collected and reviewed by the local CDC staff on the spot. Members of the study team participated in health education classes and provided on-site guidance on the implementation of school intervention measures every semester. The implementation of monthly school-based intervention measures was verified with photos and videos taken during intervention sessions. The implementation of family-based interventions was also verified with photos and screen shots. Data were double entered and cleaned with Epidata 3.1 (The Epidata Association 2008, Odense, Denmark).

### 2.10. Data Management and Statistical Analysis

Students’ height and weight, measured by school health teachers, were used to calculate body mass index (BMI), using the formula, BMI = weight (kg)/height (m)^2^. Parents’ height and weight were self-reported. China’s Adult Overweight and Obesity Prevention and Control Guidelines recommended standards were used for the cut off points of overweight and obesity: low weight (BMI < 18.5), normal weight (BMI: 18.5–23.9), overweight (BMI: 24.0–27.9) and obesity (BMI ≥ 28.0). The consumption of any SSBs at a frequency of less than once per week is classified as infrequent consumption, ≥4 times/week as frequent consumption. 

All statistical analyses were performed using the statistical software package IBM SPSS Statistics Version 20.0 (SPSS Inc., Chicago, IL, USA). Qualitative variables are presented in frequencies and percentages while quantitative variables are described as mean and standard deviation (SD). Difference in the proportions of SSBs consumption and knowledge rate of SSBs between groups by subject characteristics were compared using the chi-square test. Difference in the frequency of SSBs consumption between groups was compared using the paired samples *t*-test. A paired-sample chi-square test (McNema–Bowker test) was used to compare frequency distribution of SSBs consumption between at pre-intervention and post-intervention points within the intervention and control groups, respectively. Stepwise multivariable linear regression models are used to assess independent predictors for difference in frequency of SSBs consumption between pre-intervention and post-intervention. Students, who did not drink any SSBs during the study were excluded in the final analysis. In consideration of collinearity of dependent variables, our model includes group (intervention or control), residential area, sex, parental education level, parental BMI level, whether parents have been warned about the harms of SSBs, whether home always has SSBs, whether parents often drink SSBs and eat sugary snacks, whether parents restrict children from SSBs and sugary snacks. A *p* value below 0.05 was considered as statistically significant.

## 3. Results

### 3.1. Demographic and Clinical Characteristics

This study was conducted from September 2019 to September 2020 in Nanjing, China. 

At baseline, 1686 participants, 914 from the intervention group and 772 from the control group, were enrolled and completed the assessment. Of those who participated at baseline, 887 in the intervention group and 746 in the control group completed the trial and outcome assessment and were included in the final analysis (Figure 1). Reasons for dropping out include students’ migration to another district for school and absenteeism at school on the day of the outcome assessment. No significant differences of socio-demographic characteristics were observed between participating subjects and those dropped out (Table 1). 

Table 2 presents the social and demographic characteristics of the participants. The mean age was 9.4 years (SD ± 0.5) for 874 (53.6%) boys and 758 (46.4%) girls. Of all participants, 891 (54.6%) lived in rural areas and 742 (45.4%) in urban areas. More than half of parents had university educational attainment, and 55.1% fathers and 16.7% mothers were overweight or obese. No significant differences of social and demographic characteristics were observed between participants from the intervention group and control group.

### 3.2. Knowledge of SSBs

Prior to the intervention, 301 (33.9%) and 276 (37.0%) students in the intervention group and control group had adequate knowledge about SSBs, respectively (X^2^ = 1.66, *p* = 0.21). After intervention, 786 students (88.6%) and 400 students (53.6%) in the control group had adequate knowledge about SSBs in the intervention group. Both groups showed increasing trends in SSBs knowledge over time (intervention group X^2^ = 31.84, *p* = 0.00, control group X^2^ = 295.33, *p* = 0.00) but the intervention group had a higher proportion of students with adequate SSBs knowledge than in the control group (X^2^ = 249.60, *p* = 0.00). For each of the 10 questions about the SSBs knowledge, the proportion of students in the intervention group who answered correctly was higher than in the control group (all *p* < 0.05) after the intervention (Table 3).

### 3.3. Family Influences with Parents and at Home

At baseline, there was no significant difference in the family environment between the intervention group and control group (all *p* > 0.05). After intervention, in the intervention group, there was an increase in ‘yes’ responses to the questions on parents limiting children’s SSBs consumption (all *p* < 0.05), with 88.8% vs. 63.5% for ‘parents warned children about the harms of SSBs’, 85.2% vs. 56.6% for ‘my parents restricted me from drinking SSBs’ and 83.1% and 52.8% for ‘my parents restricted me from sugary snacks’. The ‘yes’ responses to questions on parents’ role models decreased (all *p* < 0.05), from 24.6% to 10.8% for ‘my home always have SSBs’, from 31.1% to 6.4% for ‘my parents often drink SSBs’ and from 20.5% to 4.5% for ‘my parents often eat sugary snacks’, respectively. In the control group, there was no significant difference before and after intervention in any of the family environment questions. The family environment with parents were significantly more improved in the intervention group than the control group (Table 4).

### 3.4. Frequency of SSBs Consumption

#### 3.4.1. Frequency Distribution of SSBs Consumption

At baseline, the proportion of frequent consumption of any SSBs in the intervention group was lower than that in the control group (42.5% vs. 51.2%; X^2^ = 14.01, *p* < 0.01). After intervention, students in the intervention group had significantly decreased frequent SSBs consumption from 42.5% to 31.5% (X^2^ = 61.98, *p* < 0.01) while infrequent SSBs consumption increased from 20.0% to 33.8%. Except for coffee beverages, more students consumed infrequently other types of SSBs in the intervention group, from 62.2% at baseline to 70.1% after intervention for carbonated beverages (X^2^ = 17.44, *p* < 0.01), 57.4% to 76.1% for fruit and/or vegetable beverages (X^2^ = 86.45, *p* < 0.01), from 51.7% to 68.4% for lactobacillus/milk beverages (X^2^ = 60.79, *p* < 0.01). There was no significant difference in the consumption of SSBs in the control group, the proportion of frequency SSBs consumption from was 51.2% at baseline and 56.2% at the end of the study (X^2^ = 6.90, *p* = 0.08). After intervention, the proportion of frequent consumption of SSBs in the intervention group was lower than that in the control group (31.5% vs. 56.2%; X^2^ = 114.33, *p* < 0.01) (Table 5).

#### 3.4.2. Frequency Difference in SSBs Consumption

At baseline, the weekly consumption frequency of any SSBs in the intervention group was lower than in the control group (4.55 vs. 5.24 times per week; t = −2.42, *p* = 0.02). After intervention, students in the intervention group had significantly decreased frequent SSBs consumption from 4.55 to 3.11 (mean change = −1.43, t = 6.41, *p* < 0.01). Except for plant-protein beverages and coffee beverages, students consumed other types of SSB in the intervention group, from 0.61 at baseline to 0.49 after intervention for carbonated beverages (mean change = −0.12, t = 2.46, *p* = 0.01), 0.73 to 0.39 for fruit and/or vegetable beverages (mean change = −0.33, t = 7.14, *p* < 0.01), from 1.20 to 0.78 for lactobacillus/milk beverages (mean change = −0.43, t = 5.74, *p* < 0.01). There was no significant change in the consumption of SSBs in the control group during the period. The frequency of any SSBs consumption was 5.24 per week at baseline and 5.25 at the end of the study (t = −0.04, *p* = 0.97). After intervention, the proportion of frequent consumption of SSBs in the intervention group was lower than that in the control group (3.11 vs. 5.25; t = −9.12, *p* < 0.01) (Table 6).

### 3.5. Multivariate Regression Analysis of Factors on Improvement of SSBs Intake

Table 7 describes the results of multivariate analysis. The intervention group (B = −1.77, 95% CI, −2.45 to −1.09), currently living in the urban area (B = −2.05, 95% CI, −2.77 to −1.32) and either parent’s education level being university or above (B = −0.95, 95% CI, −1.88 to −0.01) and parental education level being high school or lower (B = −1.19, 95% CI, −1.99 to −0.38), are positively associated with reduced SSBs consumption. Interventions showed an average of decrease in SSBs consumption by 1.77 units but living in the urban area showed a decrease by 2.05 units.

## 4. Discussion

To the best of our knowledge, this is the first trial combining school-based and family-based interventions to examine behavioral changes in consuming SSBs [23,24,26,31]. The combination of school-based and family-based interventions appears effective in reducing consumption of SSBs among Chinese students aged 9–10 years, as evident by the significantly reduced frequency of SSBs consumption in the intervention group during the trial. 

Schools, where children spend most of their weekdays and receive developmentally and culturally appropriate didactic lessons, present an ideal place for interactive activities to promote reduced SSBs consumption [29,33]. Parents are both role models and key decision-makers for children’s food intake [34]. Through changing the family beverage drinking environment and parental behavior of SSBs consumption, parents and other caregivers play important roles in changing children’s drinking habits [28]. Combining school-based and family-based interventions to reduce SSBs consumption is essential given the ubiquitous availability of these beverages for primary school students. Children and parents working together to reduce SSBs consumption appears to be a feasible and effective intervention strategy [33,35,36].

Our findings support the argument that schools, where children spend 8 or more hours daily during weekdays, remain one of the best settings for school children to acquire their knowledge regarding SSBs [25,26]. Despite increasing SSBs knowledge scores (88.6%) over time in both groups, the intervention group had a significant increase as compared to those (53.6%) in the control group. This is consistent with previous research that assessed children’s beverage knowledge, attitudes and habits after a multicomponent intervention. A recent review reported a multicomponent intervention (MCI) on children’s dietary diversity that improved food knowledge and preferences with a positive food environment and time to facilitate the development into healthy eating behaviors [31]. Through the intervention of health education courses, increased students’ cognition of knowledge of SSBs may impact the behavior change when consuming SSBs in the future [23,37].

The family beverage drinking environment is another important factor in reducing SSBs consumptions, as revealed in more supportive behaviors of parents in the intervention group, such as warning their children about the harms of SSBs, restricting their children from drinking SSBs. The family-based intervention measures also have a good influence on parents’ behavior. Consistent with other studies, parents not only largely decide children’s choices and intake but are foremost role models for children in their beverage choices and drinking behavior in primary school students [36,38].

Increased knowledge of parents through health lectures for parents, regular instant messages through social media such as WeChat or QQ and of children through classroom-based health education was enhanced through the Little Hands holding Big Hands initiative. Such student–parent paired collaboration enabled the continuum of school-based and family-based interventions. Previous studies suggest that factors in the home environment could predict child beverage intake, as over 80% of SSBs consumed by children are consumed in the home [28,38], and the presence of SSBs in the home is correlated with children’s SSBs consumption [29,39].

Reduction in children’s SSBs consumption in the present study was associated with the residential area and parents’ educational level. A possible explanation for this may be that people living in the urban area have better socioeconomic status compared to the suburbs [40]. Previous study identified that home food environment is associated with socioeconomic status of the parents [6,29,41]. It is of interest that the lower parental educational levels, the bigger the reduction in children’s SSBs consumption. A previous study indicated that children whose parent(s) had a relatively lower education level and/or with a low family income consumed more SSBs [31]. Parents’ practices and knowledge may play a role in determining child beverage intake [29]. 

Our findings are supported by Choon Huey Teo et al. [37], a school nutrition program, involving 523 primary schoolchildren aged 7–11 years from six selected schools in Malaysia, indicating a combination of nutrition education and a healthy school canteen environment had positive effects on eating behaviors. Our findings are also supported by Battram DS et al. [42] and Fulkerson et al. [43], where children and parents working together to reduce SSBs consumption would be an acceptable strategy for families [44]. The home-based intervention involved the entire family and targeted children’s behavioral, and environmental factors important for healthy changes in the home food environment and children’s dietary intake [36,45].

In this study, combining home-based and school-based interventions appears to be more effective in reducing consumption of SSBs in students than previous studies [26] where interventions were delivered at either home or school. Such a design may be applicable for future dietary interventions for children.

The present study has several limitations. First, the randomization was made at the primary school level rather than at the individual level. Second, the RCT design was not blinded, which could lead to social desirability or the Hawthorne effect. Third, consumption of SSBs among the children were self-reported, there may be reporting and/or recall biases, particularly among children [46]. Finally, there were no biomarker measurements. Future studies should consider the inclusion of biomarker measurement. Despite all the aforementioned limitations, the trial has a high follow-up rate, high fidelity to intervention and monitoring activities, and a large sample size. 

## 5. Conclusions

In conclusion, the results of our study show that the combination of school-based and family-based interventions could lead to a reduction in SSB consumption among primary-school students. Increasing SSBs knowledge at schools and improving the family SSB consumption environment should be targeted in healthy drinking interventions for schoolchildren, especially those living in urban areas and whose parents have a low educational level. Future research should examine whether a longer intervention or delayed post-intervention assessment would see the same level of reduction in the SSBs consumption in high school students.

## Figures and Tables

**Figure 1 nutrients-14-00833-f001:**
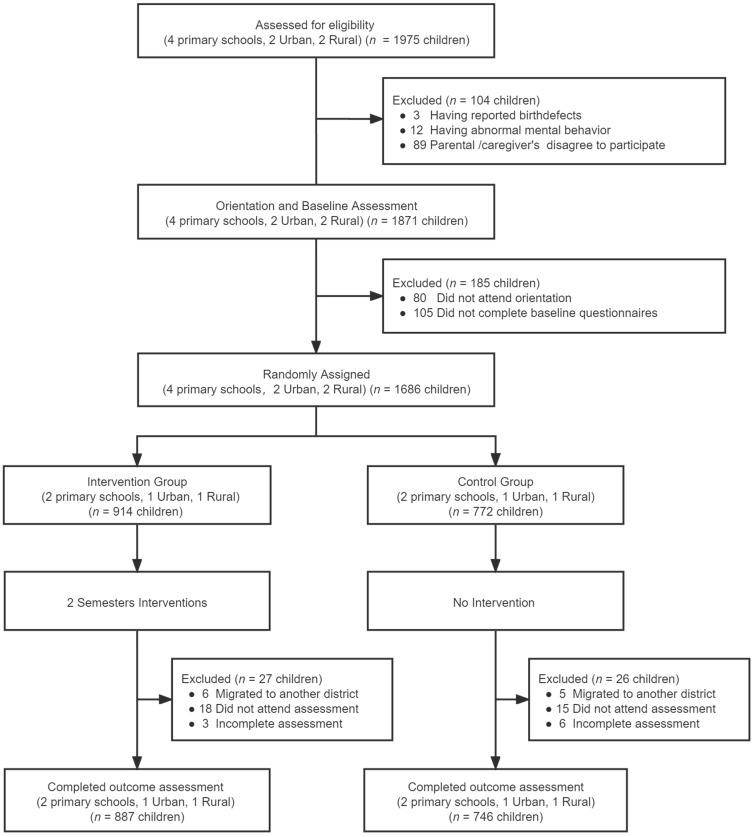
Flow of participation.

**Table 1 nutrients-14-00833-t001:** Demographic characteristics between subjects retained in trial and dropped out (N/%, Mean ± SD).

Characteristics	Retained in Trial (1633)	Dropped Out (53)	χ^2^/t	*p*
Sex				
Male	875/53.6	29/54.7	0.03	0.87
Female	758/46.4	24/45.3		
Age (years)				
9	1044/63.9	33/62.3	0.06	0.80
10	589/36.1	20/37.7		
Area				
Rural	891/54.6	31/58.5	0.32	0.57
Urban	742/45.4	22/41.5		
BMI *	17.21 ± 2.93	17.22 ± 2.96	−0.05	0.96
Frequency of SSBs consumption	4.81 ± 4.24	4.45 ± 4.48	−0.94	0.35
Father’s education level				
High school	674/41.3	23/43.4	0.10	0.76
University or above	959/58.7	30/56.6		
Mother’s education level				
High school	722/44.2	25/47.1	0.18	0.67
University or above	911/55.8	28/52.8		
Father’s BMI				
<24 kg/m^2^	733/44.9	22/41.5	0.24	0.63
≥24 kg/m^2^	900/55.1	31/58.5		
Mother’s BMI				
<24 kg/m^2^	1360/83.3	42/79.2	0.60	0.44
≥24 kg/m^2^	273/16.7	11/20.8		

* BMI, body mass index, BMI = weight (kg)/height (m)^2^.

**Table 2 nutrients-14-00833-t002:** Demographic characteristics of subjects (N/%, Mean ± SD).

Characteristics	Total (1633)	Intervention Group (887)	Control Group (746)	χ^2^/t	*p*
Sex					
Male	875/53.6	480/54.1	395/52.9	0.20	0.65
Female	758/46.4	407/45.9	351/47.1		
Age (years)					
9	1044/63.9	584/65.8	460/61.7	3.07	0.08
10	589/36.1	303/34.2	286/38.3		
Area					
Rural	891/54.6	516/58.2	375/50.3	10.22	0.00
Urban	742/45.4	371/41.8	371/49.7		
BMI	17.19 ± 2.99	17.11 ± 2.97	17.28 ± 3.01	−1.12	0.26
Frequency of SSBs consumption	4.86 ± 4.80	4.55 ± 4.99	5.24 ± 4.54	−2.42	0.02
Father’s education level					
High school	674/41.3	363/40.9	311/41.7	0.10	0.76
University or above	959/58.7	524/59.1	435/58.3		
Mother’s education level					
High school	722/44.2	395/44.5	327/43.8	0.08	0.80
University or above	911/55.8	492/55.5	419/56.2		
Father’s BMI					
<24 kg/m^2^	733/44.9	395/44.5	338/45.3	0.10	0.77
≥24 kg/m^2^	900/55.1	492/55.5	408/54.7		
Mother’s BMI					
<24 kg/m^2^	1360/83.3	724/81.6	636/85.3	3.84	0.05
≥24 kg/m^2^	273/16.7	163/18.4	110/14.7		

**Table 3 nutrients-14-00833-t003:** Knowledge awareness rate of sugar-sweetened beverages (SSBs) (N/%).

Knowledge	Before the Intervention		χ^2^	*p*	After the Intervention		χ^2^	*p*
	Intervention Group (*n* = 887)	Control Group (*n* = 746)			Intervention Group (*n* = 887)	Control Group (*n* = 746)		
1. Definition of added sugar.	183/20.6	180/24.1	2.87	0.10	618/69.7	209/28.0	281.31	0.00
2. Definition of SSBs.	190/21.4	170/22.8	0.44	0.51	593/66.9	197/26.4	265.44	0.00
3. SSBs are bad for health.	789/89.0	671/89.9	0.42	0.52	874/98.5	678/90.9	50.30	0.00
4. SSBs may cause tooth decay.	734/82.8	612/82.0	0.14	0.74	865/97.5	662/88.7	51.46	0.00
5. SSBs may cause childhood overweight and obesity.	613/69.1	542/72.7	2.46	0.13	839/94.6	582/78.0	98.51	0.00
6. SSBs can increase type 2 diabetes in children and in later life.	377/42.5	328/44.0	0.35	0.58	733/82.6	456/61.1	94.72	0.00
7. Carbonated drinks may increase risks of bone in children.	362/40.8	325/43.6	1.26	0.27	736/83.0	475/63.7	78.79	0.00
8. Fruit and/or vegetable beverage are not a substitute for fruits and vegetables.	500/56.4	425/57.0	0.06	0.84	774/87.3	500/67.0	96.75	0.00
9. Milk beverages are not substitute for milk.	305/34.4	273/36.6	0.87	0.38	613/69.1	358/48.0	74.98	0.00
10. SSBs are one of the high-sugar foods.	765/82.2	622/83.4	2.60	0.11	848/95.6	659/88.3	30.04	0.00
Adequate knowledge about SSBs *	301/33.9	276/37.0	1.66	0.21	786/88.6	400/53.6	249.60	0.00

* Correctly answered six or more items out of 10 above.

**Table 4 nutrients-14-00833-t004:** Family environment with parents and at home (N/%).

Family Environment with Parents	Before the Intervention		χ^2^	*p*	After the Intervention		χ^2^	*p*
	Intervention Group (*n* = 887)	Control Group (*n* = 746)			Intervention Group (*n* = 887)	Control Group (*n* = 746)		
Parents have been warned about harms of SSBs								
Yes	563/63.5	447/59.9	2.17	0.15	788/88.8	477/63.9	143.89	0.00
No	324/36.5	299/40.1			99/11.2	269/36.1		
My home always has SSBs ^a^								
Yes	218/24.6	163/21.8	1.69	0.20	96/10.8	167/22.4	40.10	0.00
No	669/75.4	583/78.2			791/89.2	579/77.6		
My parents restricted me from drinking SSBs ^b^								
Yes	502/56.6	422/56.6	0.00	1.00	756/85.2	424/56.8	162.98	0.00
No	385/43.4	324/43.4			131/14.8	322/43.2		
My parents restricted me from sugary snacks ^c^								
Yes	468/52.8	385/51.6	0.22	0.66	737/83.1	411/55.1	152.10	0.00
No	419/47.2	361/48.4			150/16.9	335/44.9		
My parents often drink SSBs ^d^								
Yes	276/31.1	180/24.1	9.83	0.00	57/6.4	204/27.3	132.06	0.00
No	611/68.9	566/75.9			830/93.6	542/72.7		
My parents often eat sugary snacks ^e^								
Yes	182/20.5	157/21.0	0.07	0.81	40/4.5	169/22.7	119.53	0.00
No	705/79.5	589/79.0			847/95.5	577/77.3		

^a^ my home has SSBs ≥ 4 d/week, ^b^ my parents allowed me to drink SSBs ≤ once/week, ^c^ my parents allowed me to eat sugary snacks ≤once/week, ^d^ my parents drink SSBs ≥ 4 times/week, ^e^ my parents often eat sugary snacks ≥4 times/week.

**Table 5 nutrients-14-00833-t005:** Frequency distribution of sugar-sweetened beverages (SSBs) consumption before and after intervention (N/%).

Frequency of SSBs Consumption (Times/Week)		Intervention Group (*n* = 887)		χ^2^	*p*	Control Group (*n* = 746)		χ^2^	*p*
		Before the Intervention	After the Intervention			Before the Intervention	After the Intervention		
Carbonated beverages	<1	552/62.2	622/70.1	17.44	0.00	427/57.2	408/54.7	3.03	0.388
	1~3	314/35.4	248/28.0			306/41.0	324/43.4		
	≥4	21/2.4	17/1.9			13/1.7	14/1.9		
Fruit and/or vegetable beverages	<1	509/57.4	675/76.1	86.45	0.00	374/50.1	403/54.0	4.15	0.25
	1~3	352/39.7	200/22.5			351/47.1	321/43.0		
	≥4	26/2.9	12/1.4			21/2.8	22/2.9		
Sweetened tea beverages	<1	693/78.1	731/82.4	9.81	0.02	522/70.0	520/69.7	6.57	0.09
	1~3	175/19.7	148/16.7			203/27.2	217/29.1		
	≥4	19/2.1	8/0.9			21/2.8	9/1.2		
Lactobacillus/milk beverages	<1	459/51.7	607/68.4	60.79	0.00	342/45.8	363/48.7	2.57	0.46
	1~3	338/38.1	226/25.5			316/42.4	302/40.5		
	≥4	90/10.1	54/6.1			88/11.8	81/10.9		
Sports/energy beverages	<1	605/68.2	660/74.4	12.55	0.01	459/61.5	424/56.8	5.95	0.11
	1~3	254/28.6	212/23.9			263/35.3	293/39.3		
	≥4	28/3.2	15/1.7			24/3.2	29/3.9		
Plant-protein beverages	<1	672/75.8	716/80.7	7.87	0.05	526/70.5	516/69.2	0.88	0.83
	1~3	184/20.7	142/16.0			192/25.7	196/26.3		
	≥4	31/3.5	29/3.3			28/3.8	34/4.6		
Brewed beverages	<1	764/86.1	804/90.6	14.43	0.00	602/80.7	608/81.5	1.93	0.59
	1~3	91/10.3	72/8.1			120/16.1	122/16.4		
	≥4	32/3.6	11/1.2			24/3.2	16/2.1		
Coffee beverages	<1	827/93.2	849/95.7	6.16	0.10	686/92.0	690/92.5	0.21	0.98
	1~3	55/6.2	35/3.9			51/6.8	48/6.4		
	≥4	5/0.6	3/0.3			9/1.2	8/1.1		
Any SSBs	<1	177/20.0	300/33.8	61.98	0.00	111/14.9	119/16.0	6.90	0.08
	1~3	333/37.5	308/34.7			253/33.9	208/27.9		
	≥4	377/42.5	279/31.5			382/51.2	419/56.2		

**Table 6 nutrients-14-00833-t006:** Difference in sugar-sweetened beverages (SSBs) consumption frequency (times/week) before and after intervention (Mean ± SD).

Frequency of SSBs Consumption	Intervention Group (*n* = 887)			t	*p*	Control Group (*n* = 746)			t	*p*
	Before the Intervention	After the Intervention	Mean Change			Before the Intervention	After the Intervention	Mean Change		
Carbonated beverages	0.61 ± 1.18	0.49 ± 1.01	−0.12 ± 1.39	2.46	0.01	0.62 ± 0.94	0.73 ± 1.21	0.11 ± 1.23	−2.48	0.01
Fruit and/or vegetable beverages	0.73 ± 1.19	0.39 ± 0.89	−0.33 ± 1.40	7.14	0.00	0.83 ± 1.24	0.82 ± 1.20	−0.01 ± 1.63	0.16	0.88
Sweetened tea beverages	0.42 ± 1.15	0.27 ± 0.74	−0.14 ± 1.29	3.27	0.00	0.53 ± 1.13	0.49 ± 1.08	−0.04 ± 1.35	0.87	0.38
Lactobacillus/milk beverages	1.20 ± 1.90	0.78 ± 1.55	−0.43 ± 2.20	5.74	0.00	1.34 ± 1.90	1.26 ± 1.68	−0.08 ± 2.42	0.91	0.36
Sports/energy beverages	0.58 ± 1.19	0.44 ± 1.02	−0.14 ± 1.37	2.93	0.00	0.66 ± 1.19	0.78 ± 1.30	−0.12 ± 1.56	−2.04	0.04
Plant-protein beverages	0.52 ± 1.25	0.45 ± 1.21	−0.07 ± 1.72	1.24	0.21	0.63 ± 1.43	0.67 ± 1.36	0.05 ± 1.94	−0.66	0.51
Brewed beverages	0.37 ± 1.27	0.20 ± 0.81	−0.16 ± 1.48	3.30	0.00	0.36 ± 1.18	0.28 ± 0.76	−0.11 ± 1.53	1.06	0.29
Coffee beverages	0.12 ± 0.61	0.07 ± 0.89	−0.04 ± 0.68	1.87	0.06	0.18 ± 0.82	0.16 ± 0.72	−0.02 ± 1.01	0.58	0.56
Any SSBs	4.55 ± 4.99	3.11 ± 3.22	−1.44 ± 6.65	6.41	0.00	5.24 ± 4.54	5.25 ± 4.24	0.01 ± 6.26	−0.04	0.97

**Table 7 nutrients-14-00833-t007:** Multivariate regression analysis of factors on improvement of sugar-sweetened beverages (SSBs) consumption.

Variables	Number (*n* = 1504)	Median Difference before and after Intervention ^a^ (25th, 75th Percentiles)	Β (95% CI)	*p*
Group				
Control Group	710	0.00 (−3.00, 3.00)	1	
Intervention Group	794	−1.00 (−4.00, 1.25)	−1.77 (−2.45, −1.09)	0.00
Area				
Rural	819	0.00 (−3.00, 3.00)	1	
Urban	685	−1.00 (−4.00, 2.00)	−2.05 (−2.77, −1.32)	0.00
Parental educational				
Parental education level is university or above	697	0.00 (−3.00, 3.00)	1	
One parent’s education level is university or above	286	−1.00 (−4.00, 2.00)	−0.95 (−1.88, −0.01)	0.04
Parental education level is high school or blow	521	−1.00 (−4.00, 2.00)	−1.19 (−1.99, −0.38)	0.00

^a^ Students, who did not drink any SSBs before and after the intervention, were excluded. Stepwise regression was used to adjust for confounding factors: sex (male, female), student BMI (continuous), parental BMI level (parental BMI < 24 kg/m^2^, either parent’s BMI ≥ 24 kg/m^2^, parental BMI ≥ 24 kg/m^2^), parents have been warned about the dangers of SSBs (no, yes), my home always has SSBs (no, yes), my parents often drink SSBs and eat sugary snacks (no, yes), my parents restricted me from SSBs and sugary snacks (no, yes).

## Data Availability

The data presented in this study are available from the corresponding authors on reasonable request.

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
