# Peer review of "The Combination of School-Based and Family-Based Interventions Appears Effective in Reducing the Consumption of Sugar-Sweetened Beverages, a Randomized Controlled Trial among Chinese Schoolchildren"

_nutrients, 2022, doi:10.3390/nu14040833_

Round 1
Reviewer 1 Report
This is a well-designed study to provide the effect of education of SSBs to the children and their parents.
Author Response
Dear reviewer,
Thank you very much for your time and efforts in reviewing this manuscript.
Yours sincerely,
Corresponding author: Nan Zhou

Reviewer 2 Report
This is an interesting study that looks at school and family based interventions in reducing consumption of sugar beverages among school children. The paper is well-written. I have few queries and questions regarding the study design and assumptions.
1) How did you handle confounders in the study because this is very important? Did you consider any socio-economic variables for the families enrolled in the study? Were the children in the treatment and the control group had the same level of access to these beverages? Did you consider obesity as a potential confounder? You may choose to run regression analyses and potentially filter out confounders based on risk ratios.
Take a look at this work, particularly how to filter out confounders using multi-variate regression. The methodology is relevant for your work
A. Datta et al., "‘Black Box’ to ‘Conversational’ Machine Learning: Ondansetron Reduces Risk of Hospital-Acquired Venous Thromboembolism," in IEEE Journal of Biomedical and Health Informatics, vol. 25, no. 6, pp. 2204-2214, June 2021, doi: 10.1109/JBHI.2020.3033405.
2) For parental interventions, were they done as a result of school based recommendations? I am concerned with how the two types of interventions interact with each other.
Please take a look at this paper which studies interactions between interventions on outcomes and include in the discussion how the process may be relevant to your study here since the methodology proposed in this work may be relevant to answer this question. What is the % drop in consumption when educational intervention was given alone without parental intervention and vice versa? Particluarly in the multi-variate analysis, this may potentially lead to interesting insights
Datta A, Flynn NR, Barnette DA, Woeltje KF, Miller GP, Swamidass SJ (2021) Machine learning liver-injuring drug interactions with non-steroidal anti-inflammatory drugs (NSAIDs) from a retrospective electronic health record (EHR) cohort. PLoS Comput Biol 17(7): e1009053. https://doi.org/10.1371/journal.pcbi.1009053
3) Did you consider the parents lifestyle as well before considering the groups? School children are often influenced by choices made at home by elders. I do see some results of this in table 4 and nice explanations in the discussion.
Author Response
Dear reviewers,
Thank you very much for your time and efforts in reviewing this manuscript. Those comments are very helpful for revising and improving our manuscript. We revised the manuscript with tracked changes and point-to-point responses to each of comments below.
Yours sincerely,
Nan Zhou
